# Intercomparison of Indoor Radon Measurements Under Field Conditions In the Framework of MetroRADON European Project

**DOI:** 10.3390/ijerph17051780

**Published:** 2020-03-09

**Authors:** Daniel Rabago, Ismael Fuente, Santiago Celaya, Alicia Fernandez, Enrique Fernandez, Jorge Quindos, Ricardo Pol, Giorgia Cinelli, Luis Quindos, Carlos Sainz

**Affiliations:** 1Radon Group, University of Cantabria, Santander, 39011 Cantabria, Spain; daniel.rabago@unican.es (D.R.); fuentei@unican.es (I.F.); santiago.celaya@unican.es (S.C.); alicia.fernandezv@unican.es (A.F.); enrique.fernandez@unican.es (E.F.); jorge.quindos@unican.es (J.Q.); ricpol@interface.es (R.P.); quindosl@unican.es (L.Q.); sainzc@unican.es (C.S.); 2European Commission, Joint Research Centre (JRC), I-21027 Ispra, Italy

**Keywords:** radon, proficiency test, quality assurance, metrology, interlaboratory comparison

## Abstract

Interlaboratory comparisons are a basic part of the regular quality controls of laboratories to warranty the adequate performance of test and measurements. The exercise presented in this article is the comparison of indoor radon gas measurements under field conditions performed with passive detectors and active monitors carried out in the Laboratory of Natural Radiation (LNR). The aim is to provide a direct comparison between different methodologies and to identify physical reasons for possible inconsistencies, particularly related to sampling and measurement techniques. The variation of radon concentration during the comparison showed a big range of values, with levels from approximately 0.5 to 30 kBq/m^3^. The reference values for the two exposure periods have been derived from a weighted average of participants’ results applying an iterative algorithm. The indexes used to analyze the participants’ results were the relative percentage difference *D*(%), the Zeta score (ζ), and the z-score (z). Over 80% of the results for radon in air exposure are within the interval defined by the reference value and 20% and 10% for the first and the second exposure, respectively. Most deviations were detected with the overestimating of the exposure using passive detectors due to the related degassing time of detector holder materials.

## 1. Introduction

Quality assurance is essential within the internal management of laboratories to perform tests and measurements. One of the main tools to carry out such quality control is the periodic participation in interlaboratory comparisons. On one hand, this tool affects the capacity of laboratories to carry out a specific test, and the external information obtained ensures, as far as possible, that the validation of its procedure and its internal quality control strategy are sufficiently effective with a certain degree of confidence. On the other hand, such participation includes a high potential for improvement by forcing the laboratory, given unsatisfactory results, to test its ability to detect the possible source of the error, which could be from the inadequate qualification of the staff, the incomplete validation of the test procedure, or a punctual error in the operation of the equipment, etc. [1,2].

In the field of radon, the European Council Directive 2013/59/EURATOM (EU-BSS) lays down legal limits for radon concentrations in indoor air. Therefore, to fulfill the EU-BSS requirements, it is necessary to improve the radon metrological infrastructure. Quality assurance and the traceability of in situ and laboratory measurements of radon are required. The typical situation where an intercomparison of the radon devices is developed is in a radon chamber under a radon reference atmosphere and stable environmental conditions [3]. However, the real scenarios, dwellings, and working places, where the radon monitors or passive detectors are laid can differ from the stable conditions usually performed to test them. Accordingly, some intercomparisons have been carried out in field environments in order reproduce such situations, which should be periodically conducted [4,5,6,7].

The exercise presented in this article is the comparison of indoor radon gas measurements under field conditions performed with passive detectors, giving an integrated measurement over time, and active monitors, continuously monitoring radon concentration, within the European MetroRADON project (http://metroradon.eu/). This project aims to develop reliable techniques and methodologies to enable international traceable radon activity concentration measurements and calibrations at low radon concentrations and contribute to the creation of a coordinated metrological infrastructure for radon monitoring in Europe. The specific objective of the intercomparison is to provide a direct comparison between different methodologies and to identify physical reasons for possible inconsistencies, which are particularly related to sampling and measurement techniques.

## 2. Materials and Methods 

### 2.1. Site Description

The intercomparison was carried out in the former uranium mine managed by the Spanish National Uranium Company ENUSA (Saelices el Chico, Salamanca, Spain). The reclamation of the uranium mining operations (exploited from 1972 to 2000) and the dismantling of the uranium concentrate factory started in 2001. Currently, the decommissioning activities continue restoring the affected natural space with the purpose of returning to the geological, radiological, and environmental conditions that the areas had before their exploitation began. Nowadays, the activities are mainly related with water treatment and the stability of tailings. There is acid drainage in the mine, due to the presence of pyrite in rocks of the site, which pollutes the water [8]. One of the buildings was chosen to house the Laboratory of Natural Radiation (LNR) for the calibration and testing of instruments and detectors for the measurement of natural radiation (see Figure 1) [9,10]. This place has been used to carry out interlaboratory exercises both measuring radon concentration and gamma dose rate under natural environmental fluctuations [4,5,11]. The high radioactive content in the soil along with the environmental conditions make this location a suitable place to conduct these kinds of activities.

In the LNR ground floor, there are two rooms designed as radon chambers (Room1 and Room2) with approximately 45 m^3^ volume each. Room1 has no direct connection to the exterior or any mixing system settled, while Room2 has an artificial ventilation system installed. The radon concentration homogeneity in the measurement area of Room1 is checked periodically providing statistical suitable values [9]. The radon source is the uranium mine underground soil, which has high radium content. In the east part of the LNR, a meteorological station is set up to monitor environmental conditions. 

Outside of LNR, there is a place called “Green Ballesteros”, where a 5 × 5 m^2^ and 1.5 m deep hole was dug out and filled with homogeneous soil with low radioactive content (^226^Ra concentration about 43 ± 10 Bq kg^−1^ (k = 2) and a gamma dose rate around 110 ± 5 nGy h^−1^ (k = 2) at 1 m height, k means the coverage factor) with the purpose of carrying out other kinds of intercomparisons, related with external gamma dose rate and radon in soil measurements. ^226^Ra concentration in soil was determined from 10 samples distributed in a homogenous way in the entire volume of the Green Ballesteros by the Laboratory of Environmental Radioactivity, University of Cantabria (LaRUC), which is accredited according to UNE-EN ISO/IEC 17025:2005 for activity concentration measurements of soil by gamma spectrometry using a high-purity germanium detector. The gamma dose rate was measured in situ drawn on two Reuter-Stokes devices calibrated at the Dosimetry Standards Laboratory of CIEMAT (Centre for Energy-Related, Environmental and Technological Research) [11].

### 2.2. Participants

The number of participants was 20 from 14 countries giving priority to MetroRADON partners. The list of participants is given in Table 1. There is no correlation between this table and the code assigned to each participant in the results section. 

### 2.3. Radon Exposures, Devices, and Logistic Arragements 

Radon in air exposure has been evaluated using passive detectors and active monitors inside Room1 of the LNR for two periods. The radon concentration inside Room1 is monitored remotely in order to decide when to remove the passive detectors of every exposure period. The devices were placed in Room1 on 5 December 2018 and were taken off on 6 and 8 December 2018 for the first and second exposure respectively, according to the schedule shown in Table 2.

Each participation with passive detectors required 30 units: 10 detectors for the first exposure, 10 for the second exposure, and 10 transit detectors. A total of 23 groups of passive detectors and 22 active monitors were exposed in Room1 (see Figure 2). The statistical criteria employed to select the number of exposed and transit detectors were based on the usual international intercomparison requirements [6,7,12] and with enough of a number that assured obtaining a representative average with a low uncertainty.

There were several types of passive devices used by the participants. Most of them were solid-state nuclear track detectors (SSNTD), CR-39 and LR 115, in which the detection principle is based on the microscopic defects in the detection material caused by the alpha particles of radon and its daughters that can be revealed by etching treatment [13,14]. In addition, electret ion chambers (EICs), consisting of a stable electret mounted inside an electrically conducting plastic chamber, were used for measuring radon. The negative ions produced inside the diffusion chamber by the alpha particles emitted reduce the electret’s charged surface [15]. Other procedures were implemented, e.g., using DVDs (Digital Versatile Disc) half made of polycarbonate (used as a solid state track detector) and polycarbonate foils used as a radon absorber [16,17]. The features of diffusion chambers, holders, material quality, and manufactures were diverse, too. The overall characteristics given by participants are shown in Table 3.

In this intercomparison, different active monitors were used. These kinds of electronic devices provide a time-series of radon concentration in which radon detection is based on detect alpha radiation (by ionization chamber, gross alpha counting, or alpha spectrometry) [18,19,20]. The operation modes and features are shown in Table 4. This information has been obtained from the manufacturer’s technical specifications.

The organizers placed/removed the passive detectors and active monitors from Room1. After each exposure, passive detectors were stored in a low radon concentration area that was less than 10 Bq m^−3^ determined with an AlphaGUARD (S/N EF 1763), which assures that the contribution to the total exposure is negligible. After two days, they were sealed in radon-proof aluminium bags in order to allow a proper degassing. Active monitors were turned off at the end of the second exposure. Transit detectors were stored in their original bags until the end of the second exposure. Afterwards, they were sealed in radon proof aluminium bags in order to simulate the exposed detectors conditions.

Participants have provided the exposure value and its uncertainty (k = 1) for each passive detector and the declared value for the first and second exposure period expressed in kBq m^−3^ h. In the case of active monitors, the overall exposure for each period was given; the individual radon concentration every hour was also included. 

The results provided by participants have been coded in order to maintain their anonymity. Such codification follows the rule:LxxTn
where xx is the number assigned to each participant from 01 to 20, T is the type of measurement: A for active monitor and P for passive detectors, and n is the correlative number for more than one kind of measurement group.

### 2.4. Data Analysis

The determination of the assigned value and its standard uncertainty for each radon in air exposure have been obtained by using the consensus values from participant results applying an iterative algorithm according to ISO 13528:2015 [21]. This algorithm considers the results of all participants and relocates the extreme values within the interval of acceptable deviation. 

An outlier study has been applied in order to know the extreme values. The outlier values were found from the boxplot representation and the interquartile analysis. In this case, an outlier is defined as a data point that is located 1.5 times the interquartile range (IQR) above the upper quartile and below the lower quartile. The interquartile range is defined as the difference between the third quartile (75th percentile) and the first quartile (25th percentile): IQR = (Q_3_–Q_1_). 

The robust average and robust standard deviation denoted by *E_ref_* and *s** for each radon exposure have been calculated using Algorithm 1.
**Algorithm 1.** Algorithm used to calculate the robust average *E_ref_* and robust standard deviation *s** taken from ISO 13528:2015 denoted as Algorithm A. The parameters are updated following the steps 3 and 4 iteratively until the process converges.1: There are *p* items of results denoted as:  $E_{i}=E_{1},\;E_{2},\;E_{3},\;\ldots ,\;E_{p}$2: Calculate initial values for $E_{ref}$
$\mathrm{and}\;s^{*}$ as:  $E_{ref}=\mathrm{median}\;\mathrm{of}\;E_{i}$  $s^{*}=1.483\;\mathrm{median}\;\mathrm{of}\;|E_{i}-E_{ref}|$3: Update the values of $E_{ref}$
$\mathrm{and}\;s^{*}$ as follows. Calculate:  $\delta =1.5\;s^{*}$  $E_{i}^{*}=\{\begin{array}{ccc} E_{ref}-\delta & \mathrm{when} & E_{i}<E_{ref}-\delta \\ E_{ref}+\delta & \mathrm{when} & E_{i}>E_{ref}+\delta \\ E_{i} & \mathrm{otherwise} & \end{array}$4: Calculate the new values of $E_{ref}$ and $s^{*}$ from:  $E_{ref}=\mathrm{mean}\;\mathrm{of}\;E_{i}^{*}$  $s^{*}=\;1.134\cdot \mathrm{SD}\;(E_{i}^{*}$)

Once the robust average and robust standard deviation have been calculated for each exposure period, the standard uncertainty of the assigned value is estimated as:(1)$$u\left(E_{ref}\right)=1.25\frac{s^{*}}{\sqrt{p}}.$$

The indexes used to analyze the participants’ results are the relative percentage difference *D*(%), the Zeta score (ζ), and the z-score (z).

The relative percentage difference *D*(%) has been introduced to quantify the difference between the participant’s result and the reference value obtained as consensus. Therefore:(2)$$D_{i}\left(\%\right)=100\cdot \frac{E_{i}-E_{ref}}{E_{ref}}$$
where *E_i_* is the exposure result *i* given by the participant.

The Zeta score (ζ) is a statistical index used to compare intercomparison results where the uncertainty in the measurement result is included. It is given by the following equation:(3)$$\zeta_{i}=\frac{E_{i}-E_{ref}}{\sqrt{u^{2}(E_{i})+u^{2}\left(E_{ref}\right)}}$$
in which *u*(*E_i_*) is the participant’s own estimate of the standard uncertainty of its result.

The Z-score (*z*) index is calculated as follows:(4)$$z_{i}=\frac{E_{i}-E_{ref}}{\sigma_{p}}$$
where $\sigma_{p}$ is the standard deviation for the intercomparison assessment estimated as 20% of the reference value for the first exposure and 10% of the reference value for the second one. This parameter should meet the following criterion: $u\left(E_{ref}\right)<0.3\;\sigma_{p}.$

These indexes are interpreted as follows:|ζ|; |z| ≤ 2.0 result is considered satisfactory2.0 < |ζ|; |z| < 3.0 result is considered to give a problem|ζ|; |z| ≥ 3.0 is considered not satisfactory

The Zeta score (ζ) is used together with the z-score (z) as an aid for improving the performance of participants. If a participant obtains a z-score higher than the critical value of 3.0, they may find it valuable to reassess their procedure with the subsequent uncertainty evaluation for that procedure. If the participant’s ζ score also exceeds the critical value of 3.0, it implies that the participant’s uncertainty evaluation does not include all significant sources of uncertainty. However, if a participant obtains a z-score ≥ 3.0 but a ζ score ≤ 2.0, this demonstrates that the participant may have assessed the uncertainty of their results accurately but that their results do not meet the performance expected for the proficiency testing scheme. The interpretation guidelines are shown in Table 5.

## 3. Results

Participants submitted one exposure result together with its uncertainty per group of passive detectors and/or actively monitor for the first exposure E1 and for the second exposure E2.

The variation of radon concentration in Room1 shows a big range of values, with levels from approximately 0.5 to 30 kBq m^−3^. As an example, the radon concentration measurements of the Laboratory of Environmental Radioactivity, University of Cantabria (LaRUC), taken by the calibrated device AlphaGUARD (S/N AG000032), are shown in Figure 3. The environmental conditions in Room1 during the exercise are the following: it is observed that the variation of temperature is quite stable, with an absolute difference of 1 ⁰C, the atmospheric pressure average was (935 ± 5) hPa with an absolute variation of 14 hPa, and the mean relative humidity was (63 ± 2)% with a variation range of 11%.

The assigned values used as a reference for each exposure period are derived from a weighted average of participants’ results applying the iterative algorithm described above according to ISO 13528:2015. Table 6 shows the robust average *E_ref_*, the robust standard deviation $s^{*}$, the standard uncertainty *u*(*E_ref_*), the number of results *p* and the standard deviation for the intercomparison assessment $\sigma_{p}$ estimated as 20% of reference value for the first exposure and 10% of the reference value for the second one. This parameter meets the criterion: $u\left(E_{ref}\right)<0.3\;\sigma_{p}.$

The boxplot diagram is shown in Figure 4, where the quartiles information and the outliers whose laboratory codes are displayed. There are no statistical differences between the reference exposure value calculated taking into account the total amount of results and the one calculated without considering outliers. Therefore, all the results have been considered to calculate the reference values.

Participant’s results for the first radon in air exposure are given in Figure 5. Each value is presented with its uncertainty (*k* = 1). The solid line represents the reference value obtained through consensus (356 kBq m^−3^ h), and the dashed lines denote the standard deviation for the interlaboratory assessment estimated as 20% of the reference value. Figure 6 shows the results for the second exposure, with the reference value of 1014 kBq m^−3^ h indicated with a solid line. In this case, the dashed lines represent 10% of that reference value.

About 80% of the results presented in Figure 5 and Figure 6 are within the interval defined by the exposure reference value $E_{ref}$ and the standard deviation $\sigma_{p}$ established as 20% and 10% for the first and the second exposure, respectively. The relative difference *D*(%) between each single value and the reference is shown in Figure 7. The anomalous values shown in Figure 4 are clearly out of those intervals.

Figure 8 and Figure 9 show the graphical representation of indexes used to assess the participant’s results. In some cases, the value is out of scale in order to improve the graph view. In addition, Table 7 shows the percentage of results that are within the limits for each index. For the relative difference, the percentage of results within 10% and 20% of the reference exposure in each case is presented.

The overall performance of results given by z-score is satisfactory; about 90% of the results have a value lower than 2.0 for both exposures. Only the results of three cases have a z-score value above 3.0 for the first exposure and one result for the second exposure period. Regarding the Zeta score, about 60% of results are satisfactory (|ζ| ≤ 2.0); however, 29% of results for the first exposure and 20% for the second exposure period are not satisfactory, with a Zeta score |ζ| ≥ 3.0. 

## 4. Conclusions

An interlaboratory exercise of indoor radon under field conditions has been carried out in the Laboratory of Natural Radiation (LNR) between 5 and 8 December 2018. The facility is located in the former uranium mine managed by the Spanish National Uranium Company ENUSA (Saelices el Chico, Salamanca, Spain). Radon in air measurements were assessed from two exposure periods. 

Radon in air reference values for each exposure were obtained through consensus from participant’s results applying an iterative algorithm according to ISO 13528:2015. The indexes used to analyze the participants results are the relative percentage difference *D*(%), Zeta score (*ζ*), and z-score (*z*).

Most z-score results are satisfactory; about 90% of the results have a value lower than 2 for both exposures. Only the results for three cases for the first exposure and one result for the second exposure period are not satisfactory, with z-score values higher than 3.0. 

Regarding the Zeta score, about 60% of results are satisfactory (|ζ| ≤ 2.0); however, 29% of the results for the first exposure and 20% for the second exposure period are not satisfactory, with a Zeta score |ζ| ≥ 3.0. 

Over 80% of the results for radon in air exposure are within the interval defined by the reference value and the standard deviation, which were established as 20% and 10% for the first and the second exposure, respectively. 

Five results of the first exposure are considered outliers. All of them are passive detectors and are overestimating the exposure from approximately 40% to 160%. Such deviations could be related with the degassing time of detector holder materials. Radon could get adsorbed in it for a long time, so even after two days, when the detectors were put in radon proof bags and sealed. A further difficulty in this intercomparison exercise is that the exposures are reached in a short time period with high radon concentrations in air. At the end of the first exposure period, there was a radon concentration in air around 30 kBq m^−3^ which can cause the holder degassing problem previously mentioned. In case of the second exposure, the radon concentration was under 2 kBq m^−3^ at the end of that period, therefore reducing the exposure increase due to the possible effect of adsorption and degassing.

In general, active monitors provide better results taking into account the z-score and the relative difference parameters. However, in case of the Zeta score, worse results are obtained, which can mean that the estimated uncertainty does not include all significant sources. The exercise was successful, taking into account the large number of different devices used, especially in passive detectors where the holder materials, diffusion chamber volume, detectors area, or detection principle were diverse.

## Figures and Tables

**Figure 1 ijerph-17-01780-f001:**
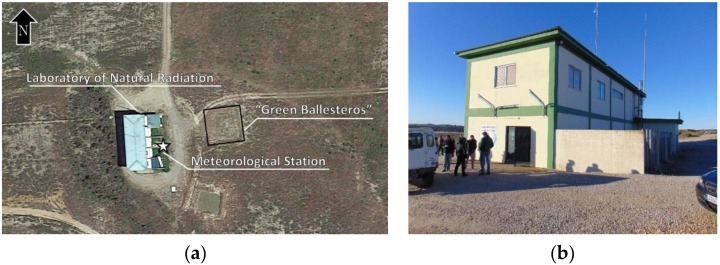
(**a**) Aerial view of the Laboratory of Natural Radiation (LNR) and surroundings where the intercomparison was developed; (**b**) Picture of LNR building taken from the south part.

**Figure 2 ijerph-17-01780-f002:**
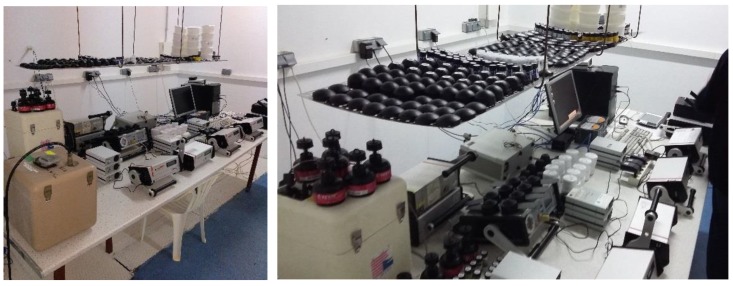
Radon devices inside Room1 (LNR: Laboratory of Natural Radiation).

**Figure 3 ijerph-17-01780-f003:**
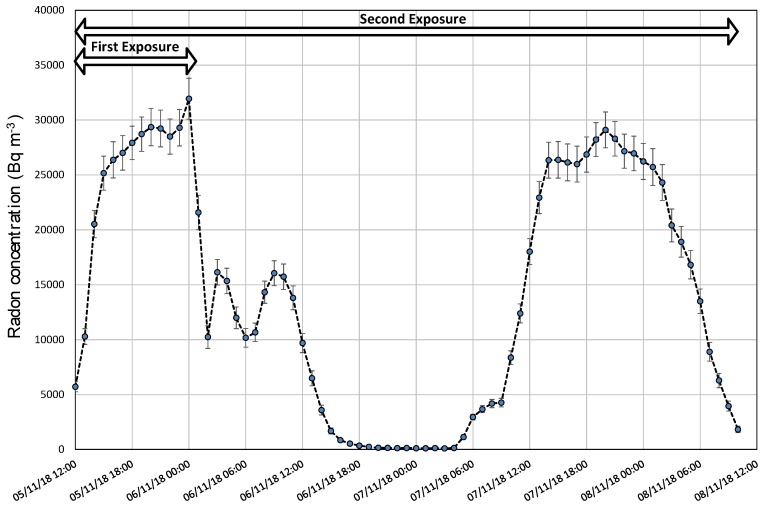
Radon concentration in Room1 during the intercomparison exercise according to LaRUC. Data is displayed every hour.

**Figure 4 ijerph-17-01780-f004:**
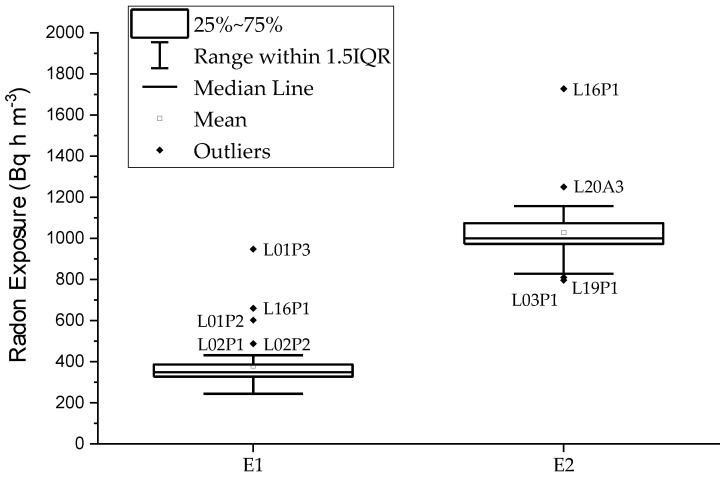
Boxplot diagram of the participant’s results for exposures E1 and E2. Outliers are identified.

**Figure 5 ijerph-17-01780-f005:**
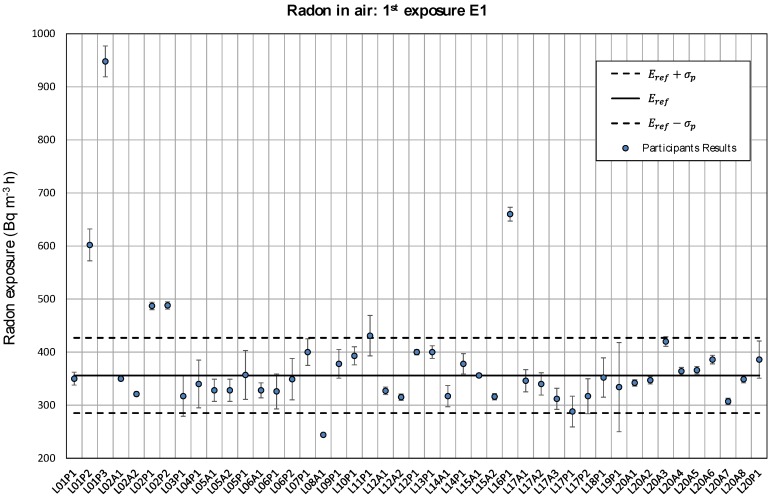
Participant’s results for the first exposure E1 with its associated uncertainty (*k* = 1). The exposure reference value is shown with a solid line and the standard deviation $\sigma_{p}=0.2E_{ref}$ with dashed lines.

**Figure 6 ijerph-17-01780-f006:**
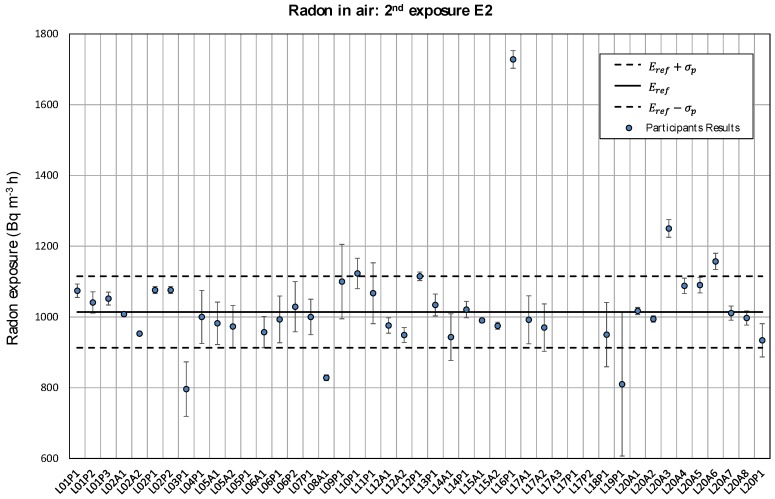
Participant’s results for the second exposure E2 with its associated uncertainty (*k* = 1). Exposure reference value is shown with a solid line and the standard deviation $\sigma_{p}=0.1E_{ref}$ with dashed lines.

**Figure 7 ijerph-17-01780-f007:**
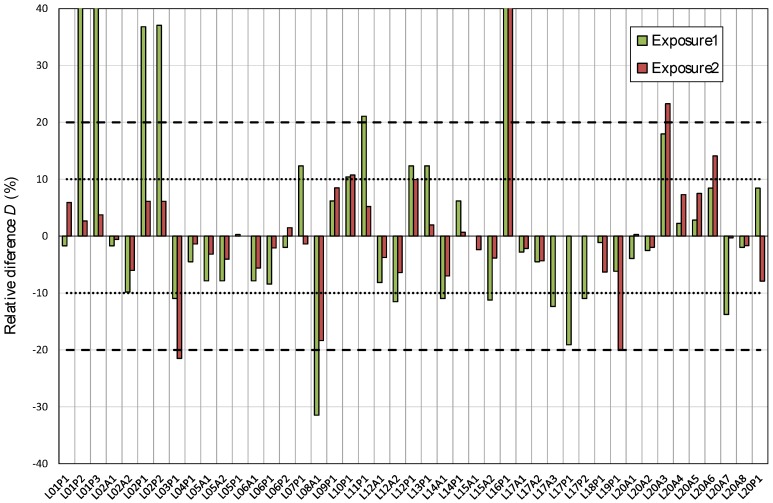
Relative difference of participant’s results to the mean value for the first and second exposure. Intervals established for the first exposure (±20%) and second exposure (±10%) are indicated. In some cases, the value is out of scale in order to improve the graph view.

**Figure 8 ijerph-17-01780-f008:**
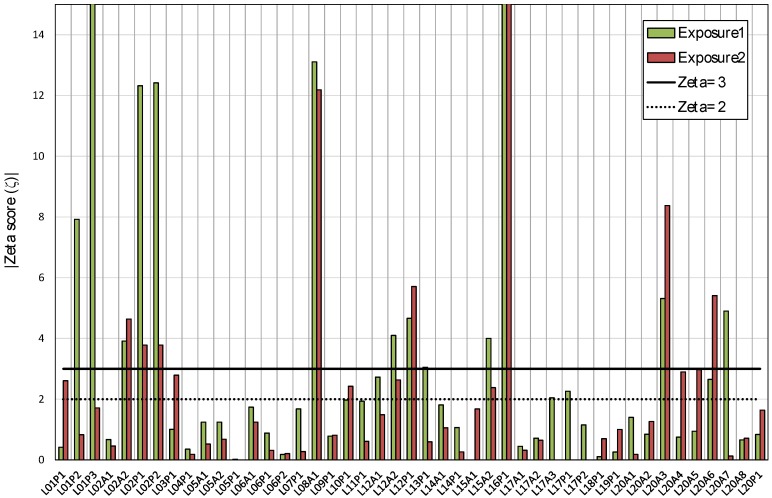
Absolute values of Zeta score for the first and second exposure.

**Figure 9 ijerph-17-01780-f009:**
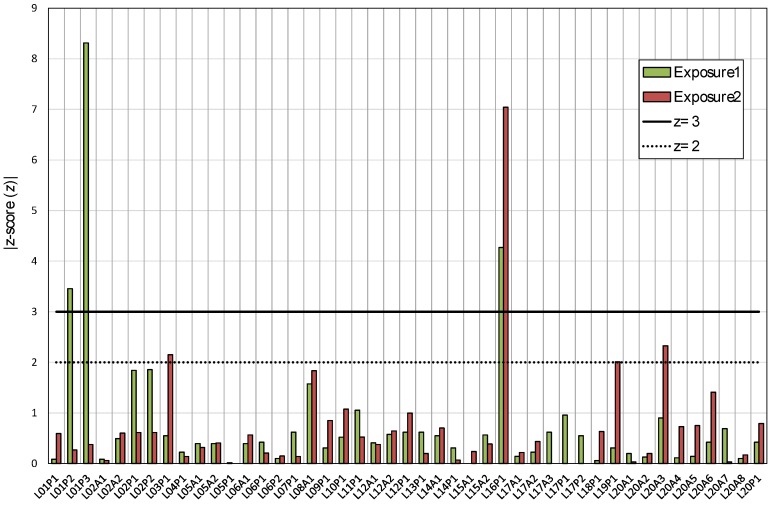
Absolute values of z-score for the first and second exposure.

**Table 1 ijerph-17-01780-t001:** Participants in the intercomparison sorted by alphabetical order.

Acronym	Institution	Country
CIEMAT	Centro de investigaciones energéticas, medioambientales y tecnológicas	Spain
CLOR	Central Laboratory for Radiological Protection	Poland
ENEA	ENEA Radon Service	Italy
INAIL	Italian National Institute for Insurance against Accidents at work	Italy
IRSN	Institut de Radioprotection et de Sûreté Nucléaire	France
JRC	Joint Research Centre	Italy
LaRUC (UC)	Laboratory of environmental radioactivity, University of Cantabria	Spain
LRAB–UEX	LRAB–Universidad de Extremadura	Spain
LRG	Laboratorio de Radón de Galicia	Spain
LRN-UC	Laboratorio de Radioatividade Natural–Universidade de Coimbra	Portugal
NRCN	Nuclear Research Center Negev	Israel
PUCP	Pontificia Universidad Católica Del Perú	Peru
RADONOVA	Radonova Laboratories AB	Sweden
Radosys	Radosys/Radosys Atlantic	Portugal/Hungary
RERA-CIEMAT	Centro de investigaciones energéticas, medioambientales y tecnológicas	Spain
STUK	Radiation and Nuclear Safety Authority	Finland
SUBG	Sofia University “St. Kliment Ohridski”	Bulgaria
SUJCHBO	National Institut for NBC Protection	Czech Republic
TR	TECNO RAD s.u.r.l.	Italy
UBB	Babes-Bolyai University	Romania

**Table 2 ijerph-17-01780-t002:** Start and end dates for each exposure ((UTC+01:00) Brussels, Copenhagen, Madrid, Paris).

	Start Date	End Date
First exposure E1:	05/11/2018 12:00	06/11/2018 1:00
Second exposure E2:	05/11/2018 12:00	08/11/2018 10:00

**Table 3 ijerph-17-01780-t003:** Passive detector features provided by the participants.

Detector	Diffusion Chamber
CR-39 RSKS 100 mm^2^ (Radosys)	Diameter 26 mm, height 55 mm 29 cm^3^ volume
CR-39 24.7 × 36.7 × 1.40 (mm) (Mi-Net)	ENEA patent
CR-39 Radout 25 × 25 × 1.5 (mm) (Mi.am)	Diameter 50 mm, height 20 mm
CR-39 TASTRAK 13 × 37 × 1 (mm) (Tasl)	Diameter 58 mm, height 20 mm NRPB/SSI
CR-39 Duotrack (Radonova)	Diameter 58 mm, height 40 mm
CR-39 Radtrak2 (Radonova)	Diameter 58 mm, height 20 mm NRPB/SSI
CR-39 Rapidos (Radonova)	Diameter 58 mm, height 40 mm
ST Electret Teflon (E-PERM)	L-OO Chamber 58 mL
ST Electret Teflon (E-PERM)	S Chamber 210 mL
LR-115 type2 400 mm^2^ (DOSIRAD)	Diameter 60.4 mm, height 27.6 mm Own design
LR-115 (KODAK) RAMARN device 0.012 mm film of cellulose nitrate, and coated on 0.1 mm thick polyester base	Polypropylene chamber 700 cm^3^ volume
Makrofol 75.7 mm^2^ STUK design “Radonpurkki”	Diameter 20 mm, height 71 mm 79 cm^3^ volume
DVD half made of polycarbonate and two thin Makrofol N foils	Thin CD case

**Table 4 ijerph-17-01780-t004:** Active monitor features used in the intercomparison.

Monitor	Detection Technology	Sensitivity (cpm at 1 kBq m^−3^)
AlphaGUARD	Ionization chamber	50
ATMOS12 DPX	Ionization chamber	20
SARAD EQF 3120	Silicon detector	7
Radon Scout	Silicon detector	1.8
Radon Scout Home	PIN photo diode	0.1

**Table 5 ijerph-17-01780-t005:** Summary of guidelines to understand the Zeta score (ζ) and z-score (*z*).

ζ Score	z-Score	Action to Take
Satisfactory	Satisfactory	Participant’s result is good. No action is required.
Not Satisfactory	Satisfactory	Participant’s claimed uncertainty is too low, but the result fulfills the intercomparison requirements.
Satisfactory	Not Satisfactory	Participant’s uncertainty assessment is accurate but the results do not fulfill the intercomparison requirements.
Not Satisfactory	Not Satisfactory	Participant’s result is biased in excess. A complete revaluation should be performed.

**Table 6 ijerph-17-01780-t006:** Reference parameters of the first exposure E1 and second exposure E2 expressed in kBq m^−3^ h obtained from participant results according to ISO 13528:2015. *p* is the dimensionless number of results.

Exposure	*E_ref_*	*u*(*E_ref_*)	$\sigma_{p}$	$s^{*}$	*p*
E1	356	8	71	43	45
E2	1014	13	101	68	41

**Table 7 ijerph-17-01780-t007:** Percentage of results that are within the limits for relative difference *D*(%), Zeta score (ζ), and z-score (z) for each exposure period classified by passive detectors and active monitors.

	Passive Detectors		Active Monitors	
	Results E1 (%)	Results E2 (%)	Results E1 (%)	Results E2 (%)
|*D*(%)| ≤ 10%	43	80	68	86
|*D*(%)| ≤ 20%	74	85	95	95
|ζ| ≤ 2.0	65	65	59	62
2.0 < |ζ| < 3.0	4	15	14	19
|ζ| ≥ 3.0	31	20	27	19
|z| ≤ 2.0	87	85	100	95
2.0 < |z| < 3.0	0	10	0	5
|z| ≥ 3.0	3	5	0	0

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
