# Peer review of "Intercomparison of Indoor Radon Measurements Under Field Conditions In the Framework of MetroRADON European Project"

_ijerph, 2020, doi:10.3390/ijerph17051780_

Round 1

Reviewer 1 Report

Interlaboratory comparisons are a basic part of the regular quality controls of laboratories to warranty the adequate performance of test and measurements. Therefore, it is necessary to master indoor radon gas measurements under field. In this paper, the author had carried out passive detectors and active monitors in the Laboratory of Natural Radiation (LNR). The topic is potentially interesting. In order to improve the level of the paper, I suggest that some Chinese research work should be referenced in the production. In addition, some reference mistakes in the manuscript should be revised, such as line 66, 83, 96…. In conclusion, it is not recommended to accept for publication after some minor revision.

Author Response

Reviewer 1

Interlaboratory comparisons are a basic part of the regular quality controls of laboratories to warranty the adequate performance of test and measurements. Therefore, it is necessary to master indoor radon gas measurements under field. In this paper, the author had carried out passive detectors and active monitors in the Laboratory of Natural Radiation (LNR). The topic is potentially interesting. In order to improve the level of the paper, I suggest that some Chinese research work should be referenced in the production. In addition, some reference mistakes in the manuscript should be revised, such as line 66, 83, 96…. In conclusion, it is not recommended to accept for publication after some minor revision.

One Chinese research reference was added about the same topic in order to support the assertion in the line 114. (Wu et al. 2014).

Reference mistakes has been solved in the entire document.

Reviewer 2 Report

Check the reference index! There are some problems in their identification in the written text.

Author Response

Check the reference index! There are some problems in their identification in the written text.

Reference mistakes has been solved in the entire document.

Reviewer 3 Report

The paper IJERPH-719977 reports results from Intercomparison radon measurements. There are several issues regarding the design and outcomes of this intercomparison, despite employing concepts of ISO 13528:2015. There are unknown parameters of the measurement site. The aspect retrieved from reading the manuscript is, more or less, that the intercomparison results determined the radon background of the site. No information on radon progeny is taken into account. The radiation protection measures are also not given. Assuming that such details are known when designing an intercomparison, it is crucial that selected results for the site are given prior to the Intercomparison design and results. The flow of text is rather descriptive and narrative. The latter should be solved. Moreover, details for average readers should be given, since the journal IJERPH has a broad audience. In this form, the manuscript seems more like a finalising report of the Intercomparison measurements, rather than a scientific paper.

The paper can be considered for publication, once significant rewording is carried out. Below some specific comments are given, that may assist the authors in the revision of their paper.

Specific Comments:

Lines 59-69: It is unclear if the restoration of the mining location is implemented and at what scale. Even so and despite referencing, this does not assures stability of radon exposure needed for inter-comparison measurements. (from, later figure 1, it known that is not stable). How is this problem solved? Especially since passive measurements of very-low duration are employed. References seem to be missing in line 66.

Line 78: How where the reported measurements taken? Which is the reproducibility? Which is the CI of the reported errors?

Line 83,91,96: Reference is missing

Lines 94-95: What are the statistical criteria employed when selecting the number of exposed and transit passive detectors?

Line 97: The reasoning is inappropriate. Please consider deleting this line.

Figure 2: Do the authors have permission from all participants to publish these two photos? Does it comply to GDPR?

Lines 101-106: Strict details and appropriate references are needed here regarding the design of detectors (additionally to Table 3). Another reference is missing here as well.

Lines 108-110. Details of the methodology of the instruments is needed for readers not familiar with these instruments.

Line 114: What was the value of this “low” concentration area. How was contamination from radon daughters on the surface of the detectors solved? What were the radiation protection measures followed?

Line 167,185: Reference is missing here.

Author Response

The paper IJERPH-719977 reports results from Intercomparison radon measurements. There are several issues regarding the design and outcomes of this intercomparison, despite employing concepts of ISO 13528:2015. There are unknown parameters of the measurement site. The aspect retrieved from reading the manuscript is, more or less, that the intercomparison results determined the radon background of the site. No information on radon progeny is taken into account. The radiation protection measures are also not given. Assuming that such details are known when designing an intercomparison, it is crucial that selected results for the site are given prior to the Intercomparison design and results. The flow of text is rather descriptive and narrative. The latter should be solved. Moreover, details for average readers should be given, since the journal IJERPH has a broad audience. In this form, the manuscript seems more like a finalising report of the Intercomparison measurements, rather than a scientific paper.

The paper can be considered for publication, once significant rewording is carried out. Below some specific comments are given, that may assist the authors in the revision of their paper.

Specific Comments:

Lines 59-69: It is unclear if the restoration of the mining location is implemented and at what scale. Even so and despite referencing, this does not assures stability of radon exposure needed for inter-comparison measurements. (from, later figure 1, it known that is not stable). How is this problem solved? Especially since passive measurements of very-low duration are employed. References seem to be missing in line 66.

The reclamation-restoration activities are currently continuing, most of them are related with water treatment and stability of the tailings. The text in the manuscript has been modified to clear this question.  A reference of the mining company was added.

This intercomparison was carried out under natural environmental radon fluctuations as it is remarked in line 72. That is the key point of this exercise. The high radon concentration allows to reach an enough radon exposure (kBq m-3 h) measured by the passive detectors in such short time. Furthermore, too much time under this high radon concentration values could saturate them.

Reference mistakes has been solved in the entire document.

Line 78: How where the reported measurements taken? Which is the reproducibility? Which is the CI of the reported errors?

The reported measurements were taken from two sources. 226Ra concentration was measured by our laboratory (LaRUC) which is accredited under UNE-EN ISO/IEC 17025 to determine the radionuclide activity concentration of soil samples by gamma spectrometry using a Germanium detector. On the other side, the gamma dose rate was measured in-situ by two Reuter-Stokes device calibrated at the Dosimetry Standards Laboratory of CIEMAT (Centre for Energy-Related, Environmental and Technological Research). More details are shown in the reference [10]. The reproducibility is assured by the historical measurements carried out in this area. The uncertainties in each case are presented with a CI=2. Text has been modified according these comments.

Line 83,91,96: Reference is missing

Solved.

Lines 94-95: What are the statistical criteria employed when selecting the number of exposed and transit passive detectors?

The statistical criteria employed to select the number of exposed and transit detectors was based on the usual international intercomparison requirements and an enough number that assure to obtain a representative average with a low uncertainty. The text in the manuscript was modified and some references were added in order to support this sentence.

Line 97: The reasoning is inappropriate. Please consider deleting this line.

Deleted.

Figure 2: Do the authors have permission from all participants to publish these two photos? Does it comply to GDPR?

The photographs are taken by one of the author. No one participant identification are showed. The previous intercomparison report, which include similar pictures, was accepted by every participant. Therefore we consider that General Data Protection Regulation is complied.

Lines 101-106: Strict details and appropriate references are needed here regarding the design of detectors (additionally to Table 3). Another reference is missing here as well.

Additional details about the detectors type and references have been added. Lines 116-125

Reference mistake solved.

Lines 108-110. Details of the methodology of the instruments is needed for readers not familiar with these instruments.

We have added some general details about the methodology and references in order to help the readers to find more information about every technique.

Line 114: What was the value of this “low” concentration area. How was contamination from radon daughters on the surface of the detectors solved? What were the radiation protection measures followed?

The radon concentration in the storage area was determined with an AlphaGUARD (S/N EF 1763). The value was less than 10 Bq/m3 that assures that the contribution to the total exposure is negligible. This information has been included in line 129.

There is not any problem with radon daughters because all the devices have filter, in case of active monitors, or diffusion chambers in case of passive detectors.

Radiation protection measures were related with trying to minimize the spent time inside the Room1, with really high radon concentration, the as minimum number of people as possible.

Line 167,185: Reference is missing here.

Solved

Reviewer 4 Report

The paper is of good quality, the results are valuable and interesting. It should be published after a minor revision.

Please check: "Error! Reference source not found.. " 

Detailed comments:

L.81: How about the homogeneity of radon concentration in Room1? Did you check it before? Is a mixing system (fan, air conditioner, etc…) installed in this room?

L.88: “Two exposures have been planned taking into account the natural radon evolution during those days”. The sentence is not clear. What does “natural radon evolution” means?

Table. 3. I propose to add the information about type of detectors/monitors calibration, i.e. primary calibration (e.g. BfS) or secondary calibration

L.114: What is the concentration of radon in the low-radon room, in which the detectors were kept for degassing after exposure,

Which device was used for the measurement

L.237: Table 1 -> Table X?

L 237: Table X. It seems to be good to distinguish of results between passive and active methods

Can authors add background and transit results provided by participants?

Author Response

The paper is of good quality, the results are valuable and interesting. It should be published after a minor revision.

Please check: "Error! Reference source not found.. " 

Solved.

Detailed comments:

L.81: How about the homogeneity of radon concentration in Room1? Did you check it before? Is a mixing system (fan, air conditioner, etc…) installed in this room?

The homogeneity is periodically checked, a reference is included where is the methodolody applied.  No mixing system settled in Room1. Manuscript text has been modified according to these comments.

L.88: “Two exposures have been planned taking into account the natural radon evolution during those days”. The sentence is not clear. What does “natural radon evolution” means?

Sentence has been clarified. The radon concentration in the Room1 is monitored remotely in order to decide when to remove the passive detectors of every exposure period.

Table. 3. I propose to add the information about type of detectors/monitors calibration, i.e. primary calibration (e.g. BfS) or secondary calibration

That information is not available. The final purpose of this kind if exercise is confirm or find issues in the calibration of each participant. Participants become to the EURADOS and MetroRADON consortium which assure a high level of quality control in every case.

L.114: What is the concentration of radon in the low-radon room, in which the detectors were kept for degassing after exposure, Which device was used for the measurement.

The radon concentration in the storage area was determined with an AlphaGUARD (S/N EF 1763). The value was less than 10 Bq/m3 that assures that the contribution to the total exposure is negligible. This information has been included in line 129.

L.237: Table 1 -> Table X?

You are right, mistake solved.

L 237: Table X. It seems to be good to distinguish of results between passive and active methods

Table 10 has been modified separating passive and active methods. Conclusions were modified in lines 298-302.

Can authors add background and transit results provided by participants?

Background and transits are not available, the information required was only the value of the every exposure in each case.  

Round 2

Reviewer 3 Report

The paper ijerph-719977 has been substantially enhanced. Despite the authors did not address radiation protection issues during the intercomparison, the introduction, methods and results are adequate for such types of experimentation. Certain protocols have been followed and the methodological steps for the analysis and interpretation of data are sufficient. The participating Laboratories were enough and from different countries.

For the above reasons, the paper is suggested for publication.